# Modulation of Structural and Physical-Chemical Properties of Fish Gelatin Hydrogel by Natural Polysaccharides

**DOI:** 10.3390/ijms26072901

**Published:** 2025-03-22

**Authors:** Aidar T. Gubaidullin, Aliya I. Galeeva, Yuriy G. Galyametdinov, Georgiy G. Ageev, Alexey A. Piryazev, Dimitri A. Ivanov, Elena A. Ermakova, Alena A. Nikiforova, Svetlana R. Derkach, Olga S. Zueva, Yuriy F. Zuev

**Affiliations:** 1Kazan Institute of Biochemistry and Biophysics, FRC Kazan Scientific Center of RAS, Lobachevsky St., 2/31, 420111 Kazan, Russiaalnikiforova22@gmail.com (A.A.N.); 2Arbuzov Institute of Organic and Physical Chemistry, FRC Kazan Scientific Center of RAS, Arbuzov Street 8, 420088 Kazan, Russia; 3Physical and Colloid Chemistry Department, Kazan National Research Technological University, 420015 Kazan, Russia; galeeva-alija@mail.ru (A.I.G.); yugal2002@mail.ru (Y.G.G.); 4Scientific Center for Genetics and Life Sciences, Sirius University of Science and Technology, Olympic Avenue, 1, 354340 Sochi, Russia; ageev.gg@talantiuspeh.ru (G.G.A.); piryazev.aa@talantiuspeh.ru (A.A.P.); ivanov.da@talantiuspeh.ru (D.A.I.); 5Institut de Sciences des Matériaux de Mulhouse–IS2M, CNRS UMR 7361, F-68057 Mulhouse, France; 6A. Butlerov Chemical Institute, Kazan Federal University, Kremlevskaya St. 18, 420008 Kazan, Russia; 7Institute of Natural Sciences and Technology, Murmansk Arctic University, Sportivnaya Str. 13, 183010 Murmansk, Russia; derkachsr@mauniver.ru; 8Institute of Electric Power Engineering and Electronics, Kazan State Power Engineering University, Krasnoselskaya St. 51, 420066 Kazan, Russia; ostefzueva@mail.ru

**Keywords:** fish gelatin, κ-carrageenan, alginate, chitosan, hydrogels, structure

## Abstract

Gelatin, a water-soluble protein, shows unique gellification properties, which determine the active commercial availability of gelatin hydrogels in modern alimentary, cosmetic, and pharmaceutical applications. The traditional sources of gelatin for industrial technologies are pork and bovine skin and bones, which sometimes produce religious and some other restrictions. In recent years, there has been a significant increase in the production of gelatin from alternative sources, such as raw fish materials. Unfortunately, fish gelatin is characterized by weak gelling ability and a decrease in gelation and melting temperature, which are a consequence of the amino acid composition and structural features of fish gelatin. One of the ways to strengthen the natural gelling properties of fish gelatin is the structural modification of gelatin hydrogels by the introduction of polysaccharides of various natural origins. We have studied the association of our laboratory-made fish gelatin with three polysaccharides, namely, κ-carrageenan, alginate, and chitosan, which have distinct chemical structures and gelling capabilities. Structural features of the studied systems were analyzed by small-angle X-ray scattering (SAXS), powder X-ray diffraction (PXRD), and scanning electron microscopy (SEM). We applied computer modeling of molecular interactions between fish gelatin and polysaccharides by means of molecular docking and molecular dynamics approaches. The existence of a correlation between the structure of gelatin-polysaccharide systems and their physicochemical properties was demonstrated by wetting angles (flow angles) and dynamic light scattering (DLS) studies of hydrodynamic sizes and surface ζ-potential.

## 1. Introduction

Gelatin, one of the most popular biopolymers, is widely used in the food, pharmaceutical, and cosmetic industries [1,2,3,4,5,6]. Gelatin hydrogels are used as a base in the development of new food and pharmaceutical products. The strength and melting point of gelatin gel are the two most commercially important functional and technological properties of gelatin [7,8,9]. These properties are determined by gel structure and the strength of intermolecular contacts in it, which, in turn, depends on molecular weight, amino acid composition, and secondary structure of gelatin, including the ratio of α and β chains [10].

In recent years, there has been a significant increase in the production of gelatin from alternative sources, such as raw fish materials [2,10,11,12,13]. One of the reasons is the religious restrictions on the use of food products of animal origin (halal and kosher products) [14]. Another important reason is related to the fact that the fishing industry produces up to 30% of waste (skin, scales, bones, and offal) in obtaining commercial products [15]. Fish gelatin is considered an excellent alternative to mammalian gelatin due to the similarity of their functional properties. However, compared with mammalian gelatin, fish gelatin has worse gelling properties, low strength, and a low melting point of gels [16], which limits its application in modern technologies. Therefore, the improvement of functional and rheological properties of fish gelatin gels is of great interest.

The use of gelatin hydrogels in different application areas poses the task of adjusting their properties in accordance with functional requirements [8,10]. One of the ways to alter the natural gelling properties of gelatin is the use of polysaccharide fillers, which can drastically improve hydrogel’s inherent properties [17,18]. To date, there are many examples of engineering the composite gelatin-polysaccharide hydrogels using gelatins in combination with κ-carrageenan [19,20], alginate [21,22,23], chitosan [24], and some other polysaccharides [25]. Electrostatic interaction and hydrogen bonding between polysaccharide and gelatin macromolecules contribute to the strengthening of the three-dimensional structural network of gel and increase its thermal stability.

The efforts of many scientific groups have shown the positive effect of polysaccharides on the rheological properties of fish gelatin hydrogels [1,11,19,21], but there is a serious lack of structural information on combined gelatin-polysaccharide systems. The objective of the current study was to study the association and structure of mixed fish gelatin-polysaccharide hydrogels, using κ-carrageenan, alginate, and chitosan, which have distinct chemical structures and gelling capabilities (Figure 1) [26,27]. The attention was focused on the correlation between the structure of gelatin-polysaccharide hydrogels and some of their physicochemical properties.

κ-Carrageenan is a linear hydrophilic sulfated galactan composed of repeating monomer disaccharide formed by 1.4-linked -D-galactose and 1.3-linked α-d-galactose [28]. The κ-carrageenan gelation takes place under the addition of monovalent or divalent cations for lowering the electrostatic repulsion between adjusting polysaccharide chains existing due to the presence of charged groups. The gelling mechanism of κ-carrageenan (Figure 1) is based on the formation of a polysaccharide secondary structure [26]. In the presence of cations, κ-carrageenan realizes a coil-to-helix transition, leading to the formation of double helices with further helix aggregation [29,30,31]. Besides a charge screening effect, the cation-bound conformation of neocarrabiose also favors the helix formation.

Alginate is built by two saccharide units, β-d-mannuronic acid (M) and α-l-guluronic acid (G) arranged in the M- and G-block regions and randomly inserted M and G units (MG blocks) [32]. One of the main alginate features is its ion-induced interaction (Figure 1), being the driving force of polysaccharide gelation [33,34,35,36]. Alginate ionotropic gels take up an intermediate position between chemical gels with the irreversible covalent chemical crosslinking of polymer chains and reversible physical gels when the polymer crosslinking is governed by electrostatics, hydrogen bonding, chain entanglement, hydrophobic interactions, and crystallization [37].

Chitosan, a product of chitin deacetylation, a naturally-derived polysaccharide, having a deacetylation degree of 50% or more, is chemically composed of 2-amino-2-deoxy-d-glycopyranose units joined by glycosidic bonds β (1 → 4) [38]. The morphology of physical chitosan hydrogels is complex and multiscale [39]. For example, chitosan physical hydrogel can be prepared by the neutralization of chitosan amino groups [40,41] under the increase of pH value. During the neutralization procedure, the NH_3_^+^ groups are deprotonated to NH_2_ due to an increase in pH, resulting in a decrease in ionic repulsion between chitosan chains, promoting chain packing and hydrogel formation. This gelling mechanism is familiar to the ion-induced gelation of alginate [42]. The morphology of such chitosan hydrogels depicts the formation of oriented capillary crystalline structures, very similar to the case of ion-induced alginate gels [39,43]. Some authors show chain ordering of chitosan molecules with a pearl necklace organization where the structure is controlled by the string between pearls [44]. Thus, one can verify the presence of local crystalline structures in the bulk of chitosan hydrogel (Figure 1), and categorize it as the crystallization gel type, similar to amylose [28].

For the gelatin gels themselves, regardless of their nature, the mechanism of gelation is based on the combined action of intermolecular (physical) interactions and crystallization [4,16,45,46,47]. The formation of composite hydrogels based on gelatin and a number of polysaccharides can lead to noticeably different systems, depending on the paired intermolecular interaction between components. Moreover, many studies, including ours, were devoted to gelatin-polysaccharide systems; unfortunately, it is not easy to compare obtained results since most of them used the gelatins and polysaccharides of different sources, structures, and properties, as the applied testing methods [4,19,21,24,43]. Here we are undertaking the idea to track the influence of several polysaccharides with different gelling natures on the structure of their combined hydrogels with fish gelatin which is the object of our systematical research [4,16,17].

The structural features of the studied systems were analyzed using small-angle X-ray scattering (SAXS), powder X-ray diffraction (PXRD), and scanning electron microscopy (SEM). We also applied the computer modeling of molecular interactions between fish gelatin and polysaccharides by means of molecular docking and molecular dynamics approaches. The existence of a correlation between the structure of gelatin-polysaccharide systems and their physicochemical properties was demonstrated by the wetting ability of samples with the measurement of marginal wetting angles (flow angles) and dynamic light scattering (DLS) studies of hydrodynamic sizes and surface ζ-potential.

## 2. Results and Discussion

### 2.1. PXRD Overview of Hydrogel Phase State

According to obtained powder X-ray diffraction experiments, all studied samples in the liquid and gel states are characterized by practically identical powder diffraction patterns, changing slightly with composition and temperature (Appendix A). It is possible to assume that the phase state of the gelatin sample does not change noticeably upon the addition of polysaccharides studied. Characteristic interatomic distances and any ordering in the X-ray scale range are close for all samples. All diffraction patterns show three broadened diffraction peaks with maxima in the regions of diffraction angles 2θ° 8.4–8.7°, 23.7–24°, and 36.4–36.8°, corresponding to the interplanar distances of 10.1–10.5Å, 3.7–3.74Å and 2.43–2.47Å, which are mainly determined by structural features of gelatin. It is generally accepted that the first peak at 10.1–10.5Å is associated with the transverse dimension of collagen-like triple helices in gelatin. The second diffuse peak characterizes the weakly ordered amorphous substances and the third one with a maximum of 2.43–2.47Å is associated with the size of repeating structural unit (pitch of helix) typical for triple helical structures [48,49,50]. Temperature transitions do not have a noticeable effect on the peak positions, and this behavior is typical for all gels.

One of the probable reasons for the weak expression of the first peak at diffraction angles 2θ° 8.4–8.7° in the studied samples with fish gelatin, may be an insignificant amount of collagen-like triple-helical fragments in comparison with mammalian gelatins because the lower content of proline and hydroxyproline amino acids [4,16,17,51]. To clarify this question, we have undertaken another sort of experiment with powder X-ray diffraction, namely, the concentrating of samples by their drying to “amplify” sol-gel transition, since it is known that gelatin is able to restore triple helical fragments during gel formation [52].

We observed (Figure 2) sol-gel transition during PXRD experiments with liquid gel samples upon their natural drying on the surface of silicon wafers, although these changes are associated not with sol-gel transition, but the sol-xerogel one due to the natural removal of water during the drying process. However, when a small amount of water was added to the xerogel back again, we still observed the swelling processes of samples and their transition to a gel-like state.

Similar real-time experiments were performed for all samples under identical conditions at room temperature. Figure 2 shows the sequentially obtained diffraction patterns of samples during the natural removal of solvent (water) and film formation. After this, a certain amount of water was added to dry films (until the swelling of the films was visually observed) and then the drying process was recorded again. For all studied samples the structural changes under water removal were reversible. The previously noted first diffraction peak at diffraction angles of 2θ° 8.4–8.7°, characterizing the collagen-like triple helical fragments, was observed more clearly, but slightly differently for studied systems. For initial fish gelatin and sample with k-carrageenan, this diffraction peak even increases in intensity during repeated moistening-drying procedures (Figure 2A,B). At the same time, in combined hydrogel with alginate, this peak was practically not observed under any conditions (Figure 2C). Similarly, for the gelatin sample with shrimp chitosan, the presence of this diffraction peak was also not detected (Figure 2D). Apparently, the presence of alginate and chitosan in composition with fish gelatin prevents the reversible rehabilitation of triple collagen-like helices.

Thus, the study of hydrogels during their natural drying on the surface of silicon wafers has shown that in the gel state, all of them are characterized by diffraction patterns in the form of two strongly broadened amorphous halos. Namely, they are peaks with maxima in the region of diffraction angles 2θ° 23.7–24° and 36.4–36.8°, corresponding to interplanar distances of 3.7–3.74 Å and 2.43–2.47 Å, which characterize the amorphous state of substances and the dimensional characteristics of repeating structural units in triple helices, respectively.

### 2.2. SAXS Structural Characterization of Studied Systems

The 1-D reduction of primary two-dimensional SAXS patterns provided SAXS curves, which were averaged over several successive experiments after subtracting background scattering from glass capillaries. Relatively high scattering intensity (Figure 3) indicates structural microheterogeneity in all samples, i.e., the presence of randomly oriented particles (zones of increased density) with characteristic dimensions in the range of the SAXS method (1–100 nm) [53,54]. In all samples the intense small-angle scattering is observed at any studied temperatures, 1 °C, 26 °C, and 45 °C (Figure 3).

The highest integral intensity is observed for the gel state of samples at 1 °C. The course of curves is rather similar, which may indicate the absence of noticeable alterations in the morphology of particles with varying temperatures. At the same time, the behavior of the κ-carrageenan-containing system arrests one’s attention, not only by highest intensity but also by temperature dynamics of scattering, which shows that the system practically retains its jelly-like structure up to 26 °C.

So, comparing composite gels with polysaccharides (Figure 4) one can see the highest integrated intensity of SAXS scattering at 1 °C (gel state) for the case of κ-carrageenan. Also, the slope of curves for this sample differs from others, indicating, probably, the largest size of particles formed in the κ-carrageenan-gelatin system in comparison with others studied, or, additionally, the presence of particle aggregation in this sample, thus indicating their high quantity and close touch. Another system with correspondingly high intensity is the composition of fish gelatin with alginate. The scattering intensities of the gelatin-chitosan system are approximately at the same level as fish gelatin alone. When the temperature of samples increases to 45 °C (sol), the systems of gelatin with κ-carrageenan and alginate exhibit the highest integrated intensity. The sample with chitosan shows the presence of a plateau in the region of ***s*** = 0.03–0.05 Å^−1^ and the lowest intensity value is seen for fish gelatin alone. According to the analysis of results by the Kratky algorithm (Figure 4C), the composite gelatin with κ-carrageenan shows the highest degree of heterogeneity in the form of a three-dimensional fractal-type structure. Probably, it is a result of regular persistent protein-polysaccharide junctions. In the second place, one can find the gelatin-alginate system, in which the organized structures are also observed but to some smaller degree. The linear form of dependence of scattering intensity at large values of vector ***s*** on the Kratky plots for the gelatin-chitosan system may be characteristic of scattering from rod-shaped structures. The shape of dependence with monotonically increasing intensity, reaching a plateau at high ***s*** values, indicates the scattering by Gaussian chains, thus demonstrating the essential structural difference of this system.

The characteristic shape of all scattering curves in a double logarithmic scale (Figure 4) indicates the possibility of particle aggregation in some cases and a certain polydispersity in the size of particles. Nevertheless, we have undertaken an attempt to analyze the morphology of hydrogels, calculating a number of structural characteristics from SAXS experimental data (some details and references are presented in Section 3.3).

The set of structural characteristics of the studied samples is presented in Table 1. Our quantitative structural data are based on the known structural algorithms for gel systems, including the ones based on gelatin and polysaccharides [55,56,57,58,59,60]. The most commonly used model with clear physical meaning is the Gauss-Lorentz gel model (G-L) [61]. The results of experimental data fitting within the framework of this model and the dimensional characteristics of the studied systems obtained are given in Table 1. Let us remember, that the Gaussian parameter (Ξ) represents the average characteristic length of static correlations in the system, while the Lorentz parameter (ξ) is the average length of dynamic correlations attributed to vibrations of polymer chains between cross-links (Figure 5).

Pure fish gelatin gel is characterized by the particles with a radius of gyration *R_g_* equal to 37–38 Å. This value corresponds to spherical particles with an effective radius of 47.8–48.8 Å. The value of fractal dimension (***d_f_*** = 2.14) is close to 2, which is characteristic of smooth surfaces of swollen polymer coils [55,57], and the increase of fractal dimension is typical for more elongated and branched networks. It is interesting to note that the particle size ***R_sp_***, calculated from its radius of gyration, corresponds to diameter 95.6–97.76 Å, which correlates quite well with static correlation length **Ξ** determined from the simulation of small-angle scattering curves in the Lorentz-Gauss gel model approximation (95.2–98.5 Å). In this case, the value of dynamic length **ξ** indicates a fairly dense cross-linking of molecular chains inside the particles. This is confirmed also by the powder diffraction data of gelatin hydrogel, indicating a small amount of collagen-like triple helical structures.

The addition of κ-carrageenan to gelatin leads to a noticeable increase in the particle radius of gyration from 37–38 Å to 51 Å. The static and dynamic parameters of the gel network within the Lorentz-Gaussian gel model change significantly (**Ξ** from 96.3 Å to 41.3 Å and ξ from 21.2 Å to 65.7 Å). The length of static correlations **Ξ** within the Lorentz-Gaussian gel model turns out to be practically comparable to the sizes of spherical particles obtained from ***R_g_***. A comparison of particle radii with the maximum distance ***D_max_*** indicates a noticeable deviation in the shape of gel particles from spherical towards the elongated globules. The calculated value of the fractal dimension of this gel is still closer in magnitude to the systems with compact structured chains, although there is a decrease of ***d_f_*** towards 1, which is a characteristic of linear structures [55].

In the case of composites of fish gelatin with alginate and chitosan a noticeable decrease in particle sizes estimated within the spherical model is observed compared to κ-carrageenan-based hydrogel. Thus, the radius of gyration ***R_g_*** of particles in the alginate system (35.9 Å) turns out to be almost equal to the ***R_g_*** of particles of the original gelatin gel (37–37.8 Å), and in the case of chitosan-based system, the particles become even noticeably smaller (22 Å). In both cases, the ***R_g_*** of particles (accordingly, their radius in the spherical approximation) and the calculated distances ***D_max_*** indicate a slight deviation of their shape from spherical.

The behavior of molecular chain parameters within the Lorentz-Gaussian gel model for fish gelatin hydrogels with alginate and chitosan turns out to be somewhat unusual.

For the alginate system the length of static correlations **Ξ** at 1 °C is 278 Å with a decrease significantly to 83 Å upon heating. For a gel with chitosan, the length of static correlations **Ξ** (and hence, the size of inhomogeneities or coils) increases with temperature growth from 174 Å to 358 Å at an almost constant value of dynamic correlation length (i.e., internal inter-chain contacts). The discrepancy between these parameters and the calculated radii of gyration of particles can be explained only by taking into account the possibility of the formation of compact particles by the polysaccharide component, which is then covered by gelatin molecules, which form their one bulk gel.

In general, it can be noted that for composites of fish gelatin with alginate and chitosan the process of segregation of individual phases containing either alginate (to a lesser extent) or chitosan (to a greater degree) is observed. Further interactions with gelatin occur probably on the surface of particles of these phases. Thus, for gel samples containing the shrimp shell chitosan or sodium alginate, a system of flexible cylindrical particles with a cross-section much smaller than their length is realized, and only for the composite with κ-carrageenan the continuous three-dimensional globular-type structures appear.

### 2.3. Molecular Modelling of Gelatin-Polysaccharide Interactions

Molecular docking was applied to analyze various variants of gelatin-polysaccharide complexes and identify the most stable ones. The molecular model of the gelatin triple helix fragment has a total positive charge equal to +6, with positively and negatively charged amino acids as well as hydrophobic residues distributed irregularly along the gelatin molecule (Figure 6). The interaction of gelatin triple helix with anionic and cationic polysaccharides was investigated using the Autodock program. The calculations revealed that negatively charged κ-carrageenan and sodium alginate interact strongly with gelatin triple helix with the preferential location of polysaccharide sulfate and acidic groups in the proximity of Arg or Lys residues of the protein. The alginate molecule with eight subunits of β-d-mannuronic acid (M) and a total charge of –8 is located along the gelatin helix axis.

In the most energetically favorable position (Figure 7), the alginate molecule connects two positively charged protein zones formed by Arg12 and Arg21 residues. The complexes are stabilized by electrostatic forces and hydrogen bonds between ligand oxygens and hydrogens of Arg21 and Arg12 residues, as well as between the galactose OH group and oxygen of Glu11 residue. In the most energetically favorable complex with gelatin, κ-carrageenan with four sulfate groups and a total charge of –4 is located in the zone enriched with positively charged residues and is placed almost perpendicular to the gelatin axis (Figure 7A). Two sulfate groups of κ-carrageenan interact via electrostatic forces with Arg8 and Lys5 residues. Hydroxyproline Hyp6 also contributes to the complex stability of forming hydrogen bonds with the OH group of galactose.

It is well known that electrostatic interactions are the main driving force in proteins and polysaccharides complexation. In the complexes of positively charged gelatin with anionic polysaccharides, the electrostatic forces are attractive and contribute to complex stability. For the interaction of positively charged chitosan with gelatin, the electrostatic forces are repulsive and give an unfavorable contribution to complex stability.

To investigate the possibility of complexation between gelatin and cationic polysaccharide, three chitosan models with different deacetylation degrees were created. One model contains eight β-*N*-acetyl-d-glucosamine residues (A8, chitin, deacetylation degree of 0%), the second model consist of eight β-d-glucosamine subunits (D8, deacetylation degree of 100%), and the third one has six β-*N*-acetyl-d-glucosamine and two β-d-glucosamine residues (A6D2, deacetylation degree of 25%). The overall charge of chitin, which contains eight *N*-acetylglucosamine subunits, is zero. In the formed complex, chitin interacts mainly with a hydrophobic zone of triple helix with residues from Gly13 to Phe23. The hydrogen bond between the Arg21 residue and the galactose OH group stabilizes the complex (Figure 7C). For chitosans, the docking procedure predicts remarkably weaker interactions with gelatin molecules (Figure 8). In the gelatin model, there are two small negatively charged zones, one is formed by three Glu11 residues of triple helix, and the second is organized by three Glu2 residues. Chitosan with two d-glucosamine and six *N*-acetyl-d-glucosamine units has a total charge equal to +2 and can form a complex by interacting with the Glu11 residues of gelatin. However, the unfavorable interaction of ligands with positively charged Arg12 and Arg8 residues of gelatin makes this complex unstable. The A6D2 molecule demonstrates favorable interaction with Glu2 residues of gelatin (Figure 7D); however, this interaction is very weak and the formed complex is not stable. The total charge of chitosan containing eight d-glucosamine units is +8. Surprisingly, this chitosan forms a complex with gelatin, located almost perpendicular to the gelatin axis in the negatively charged zone formed by Glu2 residues (Figure 7E). An increase in the density of positive charges in the D8 molecule compared with the A6D2 ligand leads to the formation of a more stable complex. The hydrogen bonds between NH_3_ groups and the oxygens of glutamic acids give additional contributions to complex stability.

The comparative analysis of molecular docking results shows that the amino acid sequence of gelatin and irregular distribution of charged and hydrophobic residues, on one the hand, allows the protein to interact with various types of ligands, including the positively and negatively charged ligands, and, on the other hand, all the complexes formed are quite loose. The free energies of complex formation are compared in Figure 8. All complexes are characterized by relatively low interaction energy (Figure 8A). While the electrostatic interactions are the driving force of gelatin-polysaccharide complex formation, the close neighboring of oppositely charged amino acid residues of gelatin prevents strong protein-ligand interactions. For all ligands, the van der Waals interactions, hydrogen bonding, and solvation effect give a remarkable contribution to total free energy (Figure 8B).

In the absence of strong electrostatic forces between molecules and the existence of zones with nonpolar residues, hydrophobic interactions, and hydrogen bonding begin to play a key role. As a consequence, the hydrophobic ligands are located in the vicinity of nonpolar residues. In contrast to charged polysaccharides, the binding site for chitin is formed by polar and hydrophobic residues with gelatin-chitin complexes stabilized by hydrogen bonds.

To check the stability of complexes predicted by molecular docking the molecular dynamics simulations [62] were performed for all obtained complexes. Calculations show that complexes formed by κ-carrageenan and alginate with gelatin were stable until the end of the simulation with RMSD of complexes not exceeding 0.5 nm. The complex of chitin with gelatin molecule was also stable for 50 ns and the RMSD of complex was in the range from 0.3 nm to 0.6 nm. At the same time, the complexes formed by two chitosan models with gelatin molecules were stable for less than 20 ns with subsequent detaching of ligands from protein.

In general, molecular docking and molecular dynamics simulations have detected two different mechanisms of gelatin-polysaccharide complex formation. For charged polysaccharides, the interaction with gelatin is determined by electrostatic forces. Anionic polysaccharides (alginate and κ–carrageenan) interact with positively charged gelatin and form stable polysaccharide-gelatin complexes. Cationic chitosan with different deacetylation degrees does not form complexes with gelatin due to unfavorable electrostatic interactions. Non-electrostatic interactions (van der Waals and hydrophobic forces) play a key role in the case of chitin, which forms a much weaker but stable complex with gelatin.

Different forms of complexes and stability can have different effects on the properties of gels formed in a combined mixture of gelatin and polysaccharides. Unfavorable electrostatic interactions between gelatin and chitosan also can influence the properties of the gel, although complexes are not formed.

### 2.4. SEM Visualization of Hydrogel Morphology

Figure 9 depicts hydrogel morphology for the initial fish gelatin system and composite systems with three polysaccharides studied. One can see that κ-carrageenan gives the most significant changes in the structure of the initial gelatin system (Figure 9B). Hydrogel net structure in the composite κ-carrageenan-gelatin system became more heterogeneous with enormously frequent inter-chain contacts penetrating all hydrogel volume. This finding correlates with PXRD experiments, which have shown the increased microheterogeneity of the κ-carrageenan-gelatin system with an increasing degree of collagen-like triple-helical content during repeated moistening and drying. The obtained SAXS results also suggest the formation of a continuous three-dimensional globular-type structure in the composite κ-carrageenan-gelatin hydrogel. Also, the molecular docking and molecular dynamics simulations have detected the formation of stable complexes of κ-carrageenan with gelatin.

Another view on the polysaccharide-gelatin morphology is seen in the case of alginate (Figure 9C). Hydrogel retains in general its regular honeycomb structure, although its regularity obtains some derangements due to the interaction of gelatin with alginate. Although molecular modeling in general has shown the formation of stable complexes alginate-gelatin, the character of polysaccharide-protein interaction is different in this case (Figure 5). The X-ray results show the presence of weak interactions of components in combined gel with alginate and even the decrease of gelatin collagen-like triple helixes.

In the case of chitosan, one can see some more disturbance of the honeycomb regularity of structure (Figure 9D) in comparison with the original fish gelatin (Figure 9A) may be due to the segregation of individual phases in this system, shown by SAXS experiments. Also, from the results of molecular modeling (Figure 7), one can see that the orientation of chitosan relative to gelatin is different in comparison with k-carrageenan and chitosan.

Summarizing the discussion on the structure of hydrogels from fish gelatin and three polysaccharides it is necessary to underline that all four studied systems, namely original fish gelatin, k-carrageenan-gelatin, alginate-gelatin, and chitosan-gelatin, are forming the regular hydrogel network with some difference of its uniformity in the micrometer scale, resulting on the interactions between protein and polysaccharide. It is also necessary to remember that some additional effect of a more dense hydrogel net for the case of κ-carrageenan may come from the twofold exceed of κ-carrageenan in composition in comparison with two other polysaccharides (see Section 3.1)

### 2.5. Some Physicochemical Properties of Hydrogels

We also provided a brief testing of studied systems on their physicochemical properties. The average size of particles in hydrogels was estimated using dynamic light scattering technique (Table 2). According to obtained SAXS data the values ***R_sph_*** ~ 5 nm of particles in gelatin hydrogel correlate fairly well with the DLS results for the gelatin sample (*D* ~ 11–13 nm). For the κ-carrageenan system, the noticeable decrease in the DLS diameter of particles is seen in comparison with its increase from X-ray experiments (Table 1). The introduction of alginate does not influence seriously the particle size, whereas, for the gelatin-chitosan system, one can see a significant increase in particle size value both from DLS and SAXS (G-L gel model) experiments. It is important to note that in any case, we detect the changes in studied hydrogel systems in dependence on polysaccharides used, which are of the same order of magnitude.

The zeta potential of particles is also provided in Table 2. One can see, that in comparison with the initial gelatin system, the most noticeable changes in the surface potential are detected in the presence of chitosan. These changes in zeta potential can be considered as the result of complex formation in studied systems.

The phase behavior and liquid crystal properties were studied using polarization optical microscopy (POM). It was found that for gelatin hydrogel and systems with κ-carrageenan and alginates the gel-like systems are formed (Figure 10), but for the chitosan-gelatin system we detected the mesomorphism with the texture characteristic for hexagonal lyotropic liquid crystal (Figure 10D). Apparently, it is the consequence of the presence of oriented capillary crystalline domains typical for chitosan gels [36].

To determine the surface energy characteristics of polymers or composite materials, the contact angle of liquid with a known surface tension is traditionally measured [63,64,65]. The study of wetting of various surfaces by test liquids is also necessary for further practical application of engineered samples. The most important factors that influence contact angle are the chemical composition of a sample, the structure of the surface and its inhomogeneities, temperature, and some other factors [66,67,68]. In this work, we present the results of wetting of studied hydrogels placed on the glass surface with water as a test liquid. When applied to glass, the coating (film) has a transparency almost comparable to the transparency of the substrate. Figure 11 shows photos of droplets.

Free surface energy, its polar and dispersion components, were determined from the measurements of contact angles of surface wetting with water and the corresponding analysis of its time dependencies in Fawkes coordinates. The work of water adhesion to the surface of films based on 1% fish gelatin is shown in Table 3. Free surface energy is an integral function of displaying the existence of polar groups in the surface layer, the conditions of surface formation, the degree of crystallinity, and the presence of modifying additives, and plays an important role in the adhesive interaction of contacting phases. The introduction of additives into fish gelatin leads to an increase of free surface energy, i.e., the energy of intermolecular interaction of particles at the interface with water. This pattern correlates with the results of wetting of modified substrates with water and the work of water surface adhesion to the films.

## 3. Materials and Methods

### 3.1. Materials

The laboratory fish gelatin (type A) was extracted from the skin of Atlantic cod following the procedure, expounded early [17]. To purify gelatin from inorganic salts, stepwise dialysis was performed. The gelatin solution was dialyzed sequentially against 5 mM EDTA solution, 10 mM phosphate buffer, and distilled water. Dialysis was finished when the electrical conductivity of dialysate became equal to the conductivity of dialyzed water. The determined average molecular weight of fish gelatin was 153 kDa (unpublished data of S.R. Derkach from Murmansk Arctic University). To prepare gelatin samples in the majority of experiments we used gelatin solutions in Milli-Q water purified with the “Arium mini” ultrapure water system (Sartorius, Gottingen, Germany). Initially required portions of gelatins were swelled in distilled water at 20 °C for 20 h, then they were stirred at 50 °C until being fully dissolved. This procedure allowed us to obtain homogeneous gelatin solutions with a pH value of 5.4.

κ-Carrageenan (product of Sigma-Aldrich, St. Louis, MO, USA) had a viscosity-average molecular weight M_η_ = 430 kDa and was used without additional purification.

Sodium alginate (A2033, Sigma-Aldrich, USA) was used as received.

The chitosan sample was obtained from shrimp shells in Murmansk Arctic University by conventional procedure of chemical deacetylation. Chitosan molecular weight is equal to 125 kDa, and the deacetylation degree to 93%.

### 3.2. Preparation of Fish Gelatin-Polysaccharide Systems

To prepare mixtures of gelatin with polysaccharides the weighed portions of fish gelatin, as well as κ-carrageenan and alginate, were filled up with Milli-Q water. A sample of chitosan was filled with a 0.1 M solution of acetic acid. All systems were left overnight for swelling. Next, aqueous dispersions of gelatin were heated and at 50 °C, the aqueous dispersions of κ-carrageenan, alginate, and chitosan were heated at 70 °C for 1 h with stirring at 1000 rpm on a JOAN Lab thermal shaker. Then, the heated solutions of polysaccharides and gelatin solutions were mixed to obtain composite solutions with a given mass ratio of polysaccharide to gelatin Z. For κ-carrageenan, Z = 0.8 was taken since we already have a collection of different data for this polysaccharide at these conditions. Unfortunately, for alginate, it was found quite difficult to dissolve it at high concentration due to its viscosity, so Z = 0.4 was chosen. For chitosan, Z was chosen to be the same as for alginate, also due to its poor solubility. The resulting aqueous mixtures were kept in an ultrasonic bath at 40 °C for 1 h, and then kept overnight at 1 °C to allow the stable gel state formation.

The main set of gelatin-polysaccharide systems contained 1 wt.% of fish gelatin 0.8 wt.% of κ-carrageenan and 0.4 wt.% of alginate or chitosan from shrimp shells. The pH values of prepared systems were 6.02, 6.29, 6.24, and 4.47 for fish gelatin and its mixture with κ-carrageenan, alginate, and chitosan, respectively. The X-ray scattering studies were fulfilled for temperatures 1 °C (gel state), 26 °C (intermediate), and 45 °C (sol state). In the SEM experiments, the samples were prepared using 2 wt.% of fish gelatin and the same Z values.

### 3.3. Small-Angle and Wide-Angle X-Ray Scattering

Small-angle X-ray scattering (SAXS) and partly Wide-angle X-ray scattering (WAXS) were performed in the Sirius University of Science and Technology, Research Center for Genetics and Life Sciences (Sochi, Russian Federation) using a combined SAXS/WAXS XeuSS diffractometer (Xenocs, Grenoble, France), (Genix3D source, Pilatus 300k two-dimensional detector, CuKα radiation, λ = 1.5418 Å). Experiments were carried out at temperatures of 1°, 26° and 45 °C. Samples in liquid state were placed in the borosilicate glass capillaries with 1.5 mm or 2 mm diameter, with low X-ray absorption coefficient. The sample-to-detector distance was 5 cm in WAXS and 70 cm in SAXS modes. These distances were calibrated using silver behenate as a calibration standard.

WAXS experiments of gel samples in real time (during their drying on the surface of a silicon wafer) were made in Distributed Spectral-Analytical Center of Shared Facilities for Study of Structure, Composition and Properties of Substances and Materials of FRC Kazan Scientific Center of RAS using the Bruker D8 Advance diffractometer, equipped with the Vario attachment and Vantec linear PSD using Cu Kα1 radiation (40 kV, 40 mA), monochromated by a curved Johansson monochromator (λ = 1.5406 Å). Data were collected in the reflection modes, liquid samples were placed on the surface of a standard silicon plate with zero diffraction, which reduces the background scattering. The samples were kept spinning (15 rpm) throughout the data collection. Patterns were recorded in the 2θ range between 3 and 90° with 0.008° steps and step time of 0.1–4.0 s. Several diffraction patterns in various experimental modes were collected for the samples. Data processing was performed using the EVA V. 11 and TOPAS V. 3 software packages [69,70].

To analyze the morphology of studied systems, a number of parameters and dimensional characteristics were calculated based on SAXS experimental data. Comparison of experimental scattering curves within the framework of various structural models, as well as their comparison with literature data, made it possible to identify several options that provide the most effective description of experimental data with an adequate and clear physical meaning. In particular, the modeling of scattering curves within the framework of a globular model with a local monodisperse environment makes it possible to determine the distribution function of distances *P*(*r*) between scattering centers in particles, which contains valuable information about the shape and size of macromolecular associates. The value of *P*(*r*) is zero if ***r*** exceeds the maximum intramolecular distance *D*_max_, which makes it possible to estimate *D*_max_ from experimental data using so-called indirect transformation methods [53,61,71,72,73]. The shape of the *P*(*r*) functions provides information about the overall shape of the particle and gives an independent way to determine *R_g_*. In this case, the radius of gyration *R_g_*, determined using the *P*(*r*) curve (Appendix A), which, in turn, is calculated from the entire experimental scattering intensity curve, turns out to be a much more stable estimation for the presence of polydisperse impurities than the *R_g_* value according to the Guinier plot [53]. *R_g_* can be used to judge the compactness of macromolecular aggregate. Thus, assuming a spherical shape of the particle, the effective average particle radius *R_sph_* can be calculated from *R_g_* using the expression *R_sph_* = √(5/3)*R_g_* [53]. Additionally, from the double logarithmic dependence of intensity on the scattering angle in the range of average values of the wave vector, one can estimate the fractal dimension of particles *d_f_*, which characterizes the “smoothness” of the particle surface. The SAXS patterns of hydrogels have also been described using the Gaussian-Lorentz gel model, which is by far the most commonly used model proposed in the literature to fit SAXS data for hydrogels over a relatively wide range of *s* [74,75]. In this case, the Gaussian parameter (Ξ) used in the latter model is the characteristic average size of static inhomogeneities and may depend on the introduction of cross-links into the system, while the Lorentz parameter (ξ) is the correlation length for polymer chains and is associated with the magnitude of interaction between vibrating polymer chains.

Data processing and calculation of structural parameters were performed using PRIMUS Version 3.3 [76], SasView Version 3.0 [77] and original software.

### 3.4. Molecular Dynamics Study of Gelatin-Polysaccharide Interactions

#### 3.4.1. Protein Modeling

The amino acid sequence for Atlantic cod collagen was taken from UniProt (accession numbers A0A8C4Z8Y5). The amino acid fragment GESGKPGRNGERGSSGPQGARGFPGTP from Gly163 to Hyp189 was used for the construction of a gelatin molecular model. The gelatin triple helix with three identical chains was built using the triple-helical collagen building script (THe BuScr) [78]. Each chain contains three Arg, one Lys, and two Glu-charged residues, with the total molecule charge equal to +6 (Figure 6).

#### 3.4.2. Polysaccharides Modeling

Three different types of polysaccharides were used in modeling: κ-carrageenan, alginate (alginic acid), and chitosan with different degrees of acetylation. The carrageenan is composed of repeating disaccharide units consisting of β-d-galactose connected with α-d-anhydrogalactose by 1–4 glycoside bonds with one sulfate group per each disaccharide. Alginate consists of blocks of (1 → 4)-linked β-d-mannuronate and α-l-guluronate residues. Chitosan is composed of randomly distributed (1 → 4)-linked β-d-glucosamine and β-*N*-acetyl-d-glucosamine. All polysaccharides consist of eight subunits and have different total charges. For chitosan, three models were created. One model contains eight β-*N*-acetyl-d-glucosamine residues, another one consists of eight β-d-glucosamine subunits and the third model has six β-*N*-acetyl-d-glucosamine and two β-d-glucosamine residues.

#### 3.4.3. Molecular Docking

Docking calculations were carried out in the AutoDock4.2 program [79]. The polysaccharide chains were considered to be flexible, and the rotation around the glycosidic bonds and the single bonds was allowed. The triple helix of gelatin was conformationally rigid. The 3D grid was constructed over the whole gelatin model. The Lamarckian genetic method was used to search for the conformational space of structural units. The interaction energies were calculated taking into account the contributions from the van der Waals and electrostatic interaction, hydrogen bonding, and the solvation effect. Fifty runs were performed for each protein-ligand system.

#### 3.4.4. Molecular Dynamics

The complexes obtained with molecular docking modeling were placed in the water box, and the corresponding number of chloride and sodium ions was added to maintain the electroneutrality and 0.15 M ion concentration. Each model was equilibrated using the typical six-step CHARMM-GUI protocol for 2 ns [80]. The all-atom molecular dynamics simulations were performed using the GROMACS 21 package [81]. The CHARMM36m force field, including the carbohydrate force field, was used for gelatin and polysaccharides [82,83]. Simulations were carried out under the NPT ensemble with periodic boundary conditions and the Particle Mesh Ewald electrostatics using a 2 fs time step. The Lennard–Jones and electrostatic interactions were calculated within a cutoff of 1.2 nm. The temperature was maintained using a Nosé–Hoover temperature coupling method with a time constant of 1 ps. For pressure coupling, a semi-isotropic Parrinello–Rahman method with a time constant of 5 ps and compressibility of 4.5 × 10^−5^ bar^−1^ was used. The pressure was maintained at 1 bar. The MD trajectories with a length of 50–100 ns were constructed for each complex, and the last 50 ns of trajectories were used for analysis. The GROMACS [81] and VMD [84] tools were used for the analysis of trajectories. To characterize conformation stability and dynamics of complexes, the next characteristics were analyzed: root mean square deviation (RMSD) of all heavy atoms of complexes, protein-ligand interaction energy, and the number of interchain hydrogen bonds (HB).

#### 3.4.5. Scanning Electron Microscopy

The morphology of freeze-dried gelatin-polysaccharide hydrogels was studied with the help of scanning electron microscopy (SEM) using the field emission scanning electron microscope “Merlin” (“Carl Zeiss”, Oberkochen, Germany). Experiments were made at an accelerating voltage of 5 kV using gelatin-polysaccharide cryogels [85] prepared as follows. The prepared gels were left overnight at 4–6 °C. Then, samples were frozen in liquid nitrogen and vacuum freeze-dried to obtain xerogels. The fractured xerogel sections were bespread with gold/palladium (80/20) for SEM studies. The experiments were carried out in the Interdisciplinary Center for Analytical Microscopy (Kazan Federal University, Kazan).

### 3.5. Physical-Chemical Studies of Gelatin-Polysaccharide Hydrogels

The hydrodynamic sizes and surface ζ-potential of systems were assessed by dynamic light scattering (DLS) with a Zetasizer Nano-ZS (Malvern Instruments Ltd., Malvern, UK) particle size and ζ-potential analyzer equipped with a helium-neon laser (633 nm, 4 mW). The graphical analysis of results was implemented using the DTS Application Software (Malvern Instruments). The ζ-potentials were measured in aqueous disperse systems (1 wt.% of fish gelatin and the same Z values) by electrophoretic light scattering according to the M3-PALS technology (rapidly and slowly alternating electric fields together with phase and frequency analysis of scattered light).

The type of liquid crystal phase and the interval of mesophase existence were determined by an Olympus BX51 polarized light microscope equipped with a Linkam precision heating system (Olympus Life Science Europa GmbH, Hamburg, Germany). The samples were heated from 20 to 70 °C at a rate of 5 °C/min. The phase transition temperatures were measured with an accuracy of ±0.1%.

The wetting ability of studied systems was determined by the measurement of marginal wetting angles (flow angles) using the sitting drop method on an Easy Drop Kruss DSA 20E device (Germany) with a sitting drop auto-dosing system in a thermostatically controlled cell at 25 ± 1 °C. The glass was used as a substrate. To determine the average value of wetting angles, three identical drops were applied to the substrate using an automatic dosing system. The accuracy of the wetting angle determination was ±0.1° at *T* = 25 ± 1 °C.

## 4. Conclusions

Gelatin, a water-soluble protein, shows unique gelling properties, which determine the active commercial availability of gelatin hydrogels in modern alimentary, cosmetic, and pharmaceutical applications. The traditional sources of gelatin for industrial technologies are pork and bovine skin and bones, which sometimes produces religious and some other restrictions. In recent years, there has been a significant increase in the production of gelatin from alternative sources, such as raw fish materials. Unfortunately, fish gelatin is characterized by weak gelling ability and a decrease in gelation and melting temperature. One of the ways to strengthen the natural gelling properties of fish gelatin is the structural modification of gelatin hydrogels by introducing polysaccharides of various natural origins.

On the basis of complementary results, it is possible to conclude that all four studied systems, namely original fish gelatin, κ-carrageenan-gelatin, alginate-gelatin, and chitosan-gelatin, form the regular hydrogel network with some difference in its uniformity on the nanoscale, resulting on the protein-polysaccharide interactions.

We have shown the increased microheterogeneity of κ-carrageenan-gelatin with the growth of the degree of collagen-like triple-helical content in the gel state. Our results also suggest the formation of a continuous globular-type structure in the composite κ-carrageenan-gelatin hydrogel. This system shows the highest degree of heterogeneity in the form of a three-dimensional fractal-type structure.

Some distinct picture of polysaccharide-gelatin morphology is seen in the case of alginate. Hydrogel retains in general its regular structure, although its regularity obtained some derangements due to the interaction of gelatin with alginate. Although molecular modeling has shown in general the formation of stable complexes alginate-gelatin, the character of polysaccharide-protein interaction is different in this case. The X-ray results show some weaker interactions of components in combined gel with alginate and even the decrease of gelatin collagen-like triple helixes.

In the case of chitosan, one can see some more disturbance of the honeycomb regularity of structure in comparison with the original fish gelatin may be due to the segregation of individual phases in this system and the formation of rod-shaped structures, as shown by SAXS experiments. The obtained data suggests the possibility of the formation of compact chitosan particles, then covered by gelatin molecules which form one bulk gel.

The obtained results of this study show that one can use different polysaccharides for the management and conductance of different design schemes for molecular engineering of the structure and properties of gelatin-based systems. The obtained results may be useful for the development of gel-like edible food films with high mechanical and barrier properties. The gelling potential of polysaccharide-modified gelatin can also be used in the production of pharmaceutical capsules for the delivery of biologically active ingredients and 3D-printed products.

## Figures and Tables

**Figure 1 ijms-26-02901-f001:**
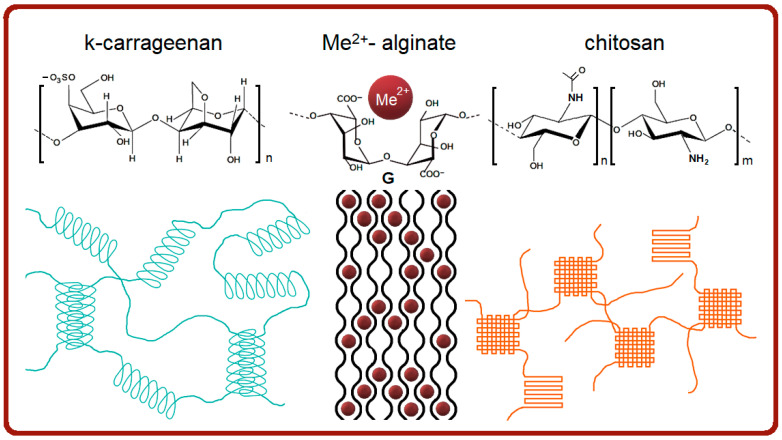
Chemical structure (**top**) and jelling schemes (**bottom**) of κ-carrageenan, alginate, and chitosan.

**Figure 2 ijms-26-02901-f002:**
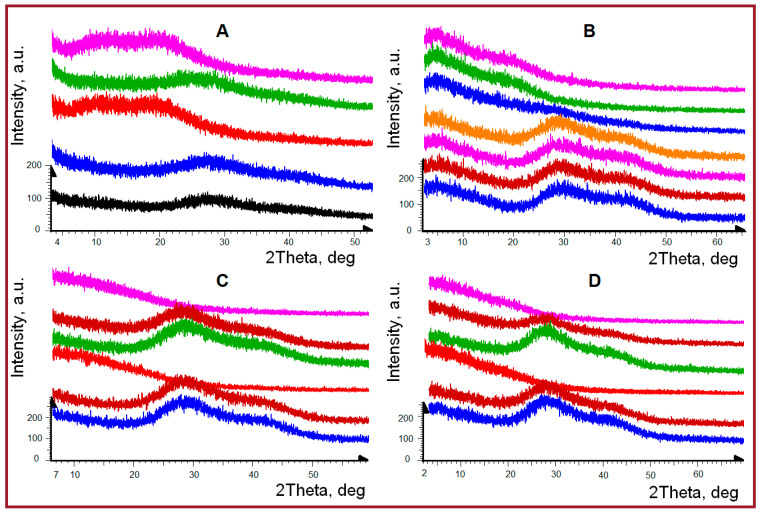
Experimental PXRD curves obtained during natural drying of samples on silicon wafer (bottom-up) for fish gelatin (black—initial in a series of experiments, red—final after drying, green—after repeated swelling, magenta—drying after repeated swelling (**A**); dynamics of gel drying for combined systems of κ-carrageenan-gelatin (**B**); alginate-gelatin (**C**) and chitosan-gelatin (**D**) in process of gel drying, rewetting of film and new drying. For clarity, the curves are shifted along the intensity axis.

**Figure 3 ijms-26-02901-f003:**
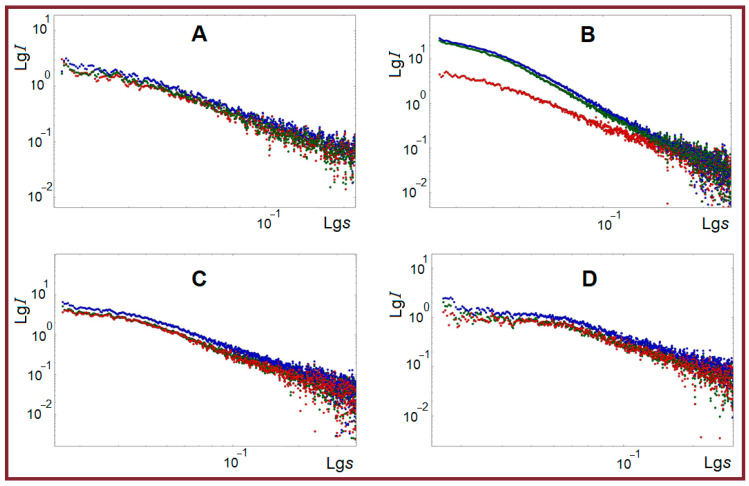
Experimental SAXS curves in double logarithmic scale for pure fish gelatin (**A**), combined gelatin systems with κ-carrageenan (**B**), alginate (**C**), and chitosan (**D**) for three temperatures: 1 °C (blue), 26 °C (green) and 45 °C (red). Scattering vector *s* = 4πsinθ/λ, Å^−1^; λ = 1.5418 Å is the X-ray wavelength.

**Figure 4 ijms-26-02901-f004:**
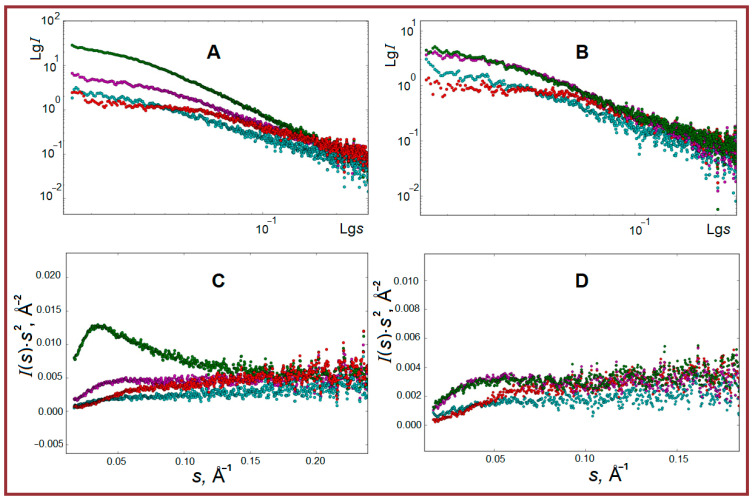
Experimental SAXS curves in double logarithmic scale at 1 °C (**A**), 45 °C (**B**), and Kratky plot at 1 °C (**C**), 45 °C (**D**) for pure fish gelatin (cyan), combined gelatin systems with κ-carrageenan (green), alginate (magenta) and chitosan (red). Scattering vector *s* = 4πsinθ/λ, Å^−1^; λ = 1.5418 Å is the X-ray wavelength.

**Figure 5 ijms-26-02901-f005:**
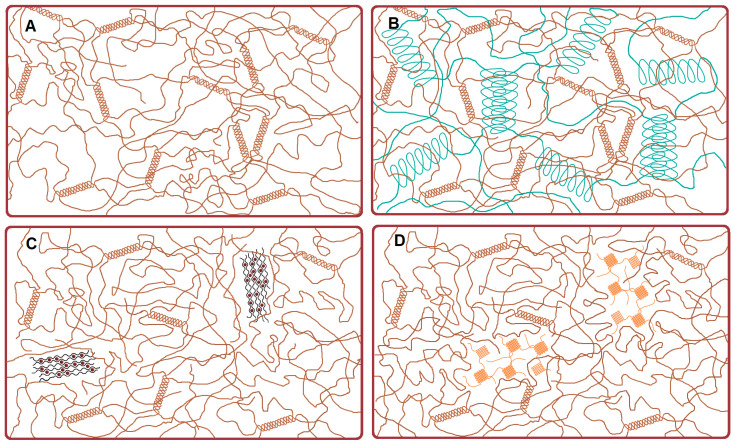
Hypothetical schemes of hydrogel morphological structure: (**A**)—gelatin, (**B**)—gelatin/κ-carrageenan, (**C**)—gelatin/alginate, (**D**)—gelatin/chitosan.

**Figure 6 ijms-26-02901-f006:**
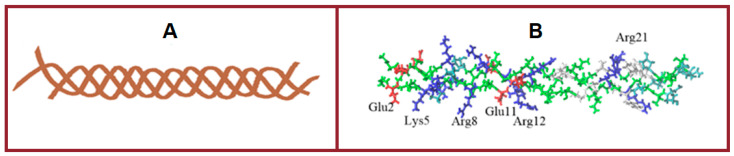
Structure of model gelatin fragment: (**A**)—gelatin triple helix; (**B**)—residues are colored according to residue type: positively charged—blue, negatively charged—red, polar—green, and hydrophobic—grey.

**Figure 7 ijms-26-02901-f007:**
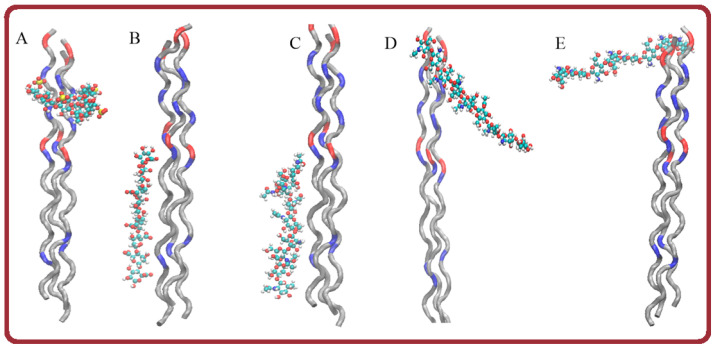
The structures of complexes formed by gelatin molecule with κ-carrageenan (**A**), alginate (**B**), chitin (**C**), chitosan with two N-acetylglucosamine subunits (**D**) and chitosan with eight N-acetylglucosamine subunits (**E**). Gelatin is presented as ribbons, positively charged residues are colored in blue, negatively charged—in red, other residues are in grey, ligands are shown as balls and sticks, colored by atom names (protons—white, carbons—cyan, oxygens—red and nitrogens—blue).

**Figure 8 ijms-26-02901-f008:**
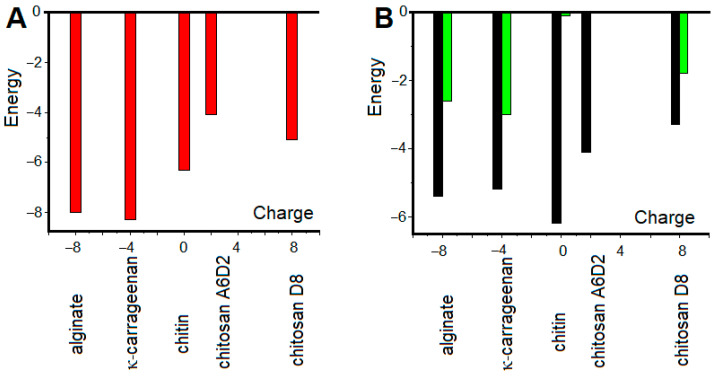
Total polysaccharide-gelatin free energy (**A**), and its electrostatic (green columns) and non-electrostatic (black columns) components (**B**) (in kcal/mol).

**Figure 9 ijms-26-02901-f009:**
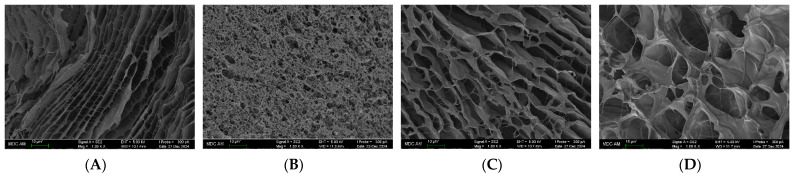
SEM images of xerogels: (**A**)—fish gelatin (FG), (**B**)—FG/κ-carrageenan, (**C**)—FG/alginate, (**D**)—FG/chitosan. Green scale bar is 10 μM for all images.

**Figure 10 ijms-26-02901-f010:**
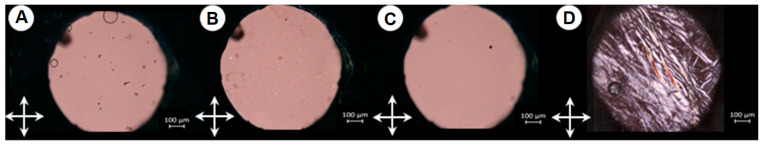
Microphotographs in polarized light of: 10% gelatin (**A**), 10% gelatin/carrageenan (**B**), 10% gelatin/sodium alginate (**C**), 10% gelatin/shrimp chitosan (**D**), (×100).

**Figure 11 ijms-26-02901-f011:**
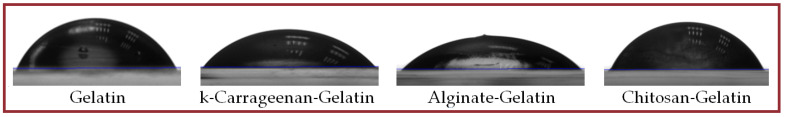
Photos of a sedentary drop of test liquid (water) for 1% fish gelatin and polysaccharide-gelatin film samples.

**Table 1 ijms-26-02901-t001:** Structural parameters and fractal dimension ***d_f_*** for fish gelatin and gelatin-polysaccharide gels (T = 1 °C and 45 °C).

	Structural Parameter, Å	T, °C	Gelatin	κ-Carrageenan- Gelatin-	Alginate- Gelatin-	Chitosan- Gelatin-
*Sphere*	*R_g_*	1	37.5 ± 2.0	50.6 ± 0.3	35.9 ± 0.7	22.0 ± 1.4
		45	37.8 ± 2.3	41.8 ± 1.1	36.1 ± 1.0	20.2 ± 1.8
	*R_sph_* $=R_{g}\sqrt{5/3}$	1	48.4	65.3	46.4	28.4
		45	48.8	53.9	46.6	26.1
	*D* _max_	1	119.05	192.18	107.56	71.9
		45	133.86	138.00	37.91	32.6
*G-L-gel- model*	Ξ, *static*	1	96.3 ± 5.7	41.3 ± 1.1	278.9 ± 8.4	174.1 ± 7.9
		45	98.5 ± 6.8	50.3 ± 3.3	83.0 ± 4.2	358 ± 27
	ξ, *dynamic*	1	21.2 ± 1.6	65.7 ± 1.1	35.3 ± 0.8	16.4 ± 1.2
		45	20.9 ± 2.1	24.3 ± 1.8	130.8 ± 4.6	15.8 ± 1.6
*Fractal dimension*	*d_f_*	1	2.14	1.62	1.18	1.00
		45	1.84	1.33	2.74	2.6

**Table 2 ijms-26-02901-t002:** Hydrodynamic diameter and ζ-potential of particles in hydrogels.

Sample	*D*, nm	ζ, mV
1% gelatin	11.3	−4.09
10% gelatin	12.0	
1% gelatin/k-carrageenan	4.5	−2.00
1% gelatin/alginate	14.6	−6.63
1% gelatin/chitosan	54.7	15.5

**Table 3 ijms-26-02901-t003:** Contact angle and work of water surface adhesion for 1% fish gelatin and polysaccharide-gelatin compositions.

	Gelatin	k-Carrageenan-Gelatin	Alginate-Gelatin	Chitosan-Gelatin
Contact angle, °	64.8	47.4	42.9	63.5
Work of surface adhesion, mJ/m^2^	102.9	121.1	125.1	104.4

## Data Availability

The data in this study are available on reasonable request from the corresponding author.

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
