# Peer review of "Modulation of Structural and Physical-Chemical Properties of Fish Gelatin Hydrogel by Natural Polysaccharides"

_ijms, 2025, doi:10.3390/ijms26072901_

Round 1
Reviewer 1 Report
Comments and Suggestions for Authors
This manuscript explores the modulation of structural and physicochemical properties of fish gelatin hydrogels through the incorporation of natural polysaccharides, using advanced experimental techniques and molecular modeling to establish molecular-level correlations.
The introduction is repetitive and certain technical sections, such as molecular modeling, contain exessive detail that could be summarized without losing essential information. Furthermore, the manuscript suffers from overly long and unclear sentences, making it difficult to follow, with key ideas burried in lengthy paragraphs. While the scientific content is solid, with a rigorous quantitative analysis using advanced experimental and theoretical techniques, the article is excesively long, and it would be desirable to shorten it to improve readability and conciseness. A more structured and streamlined presentation would enhance clarity and make the findings more accessible.
Regarding the scientific content of the manuscript, I find it to be quite robust, encompassing a comprehensive analysis of the systems studed. The work effectively combines advanced experimental techniques (with theoretical methods, including molecular modeling, docking, and molecular dynamics simulations. This approach allows for detailed characterization of the interactions between fish gelatin and polysaccharides (κ‑carrageenan, alginate, and chitosan) and establishes clear correlatons between the molecular-level structure and the physicochemical properties of the hydrogels. The data obtained are consistant and presented with quantitative rigor, which supports the conclusions regarding the gelling mechanisms and the stability of the formed complexes. Additionally, the use of multiple approaches to address the same issue provides a more comprehensive and reliable view of the behavior of these materials, which is particularly relevant for their applications in fields such as food, cosmetics, and pharmaceuticals.
However, although the scientific part is solid, I recommend improving the clarity and organization of the text to facilitate understanding of the findings and the logical flow of the argument. In summary, the manuscript makes valuable contributions to the knowledge of modified gelatin hydrogels and demonstrates a high level of experimental and analytical rigor, but it would benefit from both a reduction in length and minor corrections in language and formatting.
Comments on the Quality of English LanguageA major revision of the manuscript is necessary. The manuscript presents several issues related to language, scientific terminology, and structure. The English phrasing is often unnatural, with forced grammatical constructions, redundancies, and fluency problems. There are also grammatical and syntactic errors, such as the incorrect use of articles (e.g., "the water-soluble protein" should be "a water-soluble protein") and confusing sentence structures (e.g., "One of the reasons the religious restrictions..." is grammatically incorrect). Additionally, some scientific terms are misused, such as "jelling" instead of the correct "gelling" and "physical-chemical properties" instead of "physicochemical properties." Please revise deeply.
Author Response
The authors would like to thank Dear Reviewer for his valuable suggestions.
We took into account all the comments and made the appropriate corrections to the article. A major revision of the manuscript text is made. The main changes in the text are given in green. We made serious changes in the Introduction and some technical sections, such as molecular modeling to remove the excessive details without losing essential information. We also tried to correct the overly long and unclear sentences. We also checked our English and made corresponding corrections.
We hope that these corrections will lead to the improvement of manuscript.
Reviewer 2 Report
Comments and Suggestions for Authors
Reviewer report for Manuscript ID ijms-3520270
In the manuscript submitted for review, “Modulation of structural and physical-
chemical properties of fish gelatin hydrogel by natural polysaccharides” describes the results
for gelatin and the modifications proposed by the manuscript's authors. The authors studied the
behavior of gelatin in combination with three polysaccharides. SAXS, PXRD and morphology
analysis by SEM were performed. Molecular studies were also performed.
After reading the entire text, it is hard to see for what purpose the studies were actually
performed. The manuscript lacks a clearly stated purpose of the research: why? why? for what
purpose? What application?
The authors state the possible field of application in a very general way in the “Conclusion”
section (in one sentence).
1. I recommend to clarify the purpose of the research, the result and the applicability.
2. I suggest including a graphic of the materials produced in the manuscript.
3. In the Materials and Methods section, add a graphic/schematic showing the process of
producing the material. This will be more readable for future audiences.
4. What is the proline responsible for and what is the hydroxyproline responsible for in the
polymer structure? How does this relate to the mechanical characteristics of gelatin (line 57-
72)?
5. line 185 - 188. the authors quite boldly, actually without providing proof, suggest
“Apparently” that the presence of alginate and chitosan in the composition with fish gelatin
prevents the reversible rehabilitation of collagen-like triple helixes. On what basis was such a
conclusion made?
6. line 278-187. reference should be made to more similar results, which will give more
credence to the authors' results obtained.
7. The section 2.4 SEM Visualization of Hydrogel Morphology, line 477-484. can the authors
refer to the influence of the structure of the different materials on the applicability?
8. line 553-555. the authors report that the content of K+, Ca2+ and Na+ ions in the
polysaccharide did not exceed 6, 1 and 1% by weight, respectively. What effect on the
polysaccharide can increasing the %wages of these ions have?
9. why were the studies described in Section 3.3 Small-Angle and Wide-Angle X-ray Scattering
performed at 1°, 26° and 45°C?
10. Complete the literature list with missing DOI numbers.
For each graphic, please indicate the source - if it is not prepared by the authors of the
manuscript. What programs and licenses were used to prepare the graphics included in the
manuscript? To be completed.

Author Response
The authors would like to thank Dear Reviewer for his valuable suggestions.
We took into account the comments and made the appropriate corrections to the article. A major revision of the manuscript text is made. The main changes in the text are given in green.
1.I recommend to clarify the purpose of the research, the result and the applicability.
Answer: We made serious changes in the Introduction and add some information to the Summary. We tried to clarify the purpose of research and the applicability of results.
- I suggest including a graphic of the materials produced in the manuscript.
- In the Materials and Methods section, add a graphic/schematic showing the process of
producing the material. This will be more readable for future audiences.
Answer to points 2 and 3:
Dear Reviewer, thank you for these suggestions. We suppose that such graphical information is not necessary in this work. The procedure of samples preparing is traditional and rather simple, explained clearly in the text. It is not the first our article in the field of gelatin/polysaccharide hydrogels and we are using such form of presentation if it is not necessary to accent the readers attention on some special details.
- What is the proline responsible for and what is the hydroxyproline responsible for in the
polymer structure? How does this relate to the mechanical characteristics of gelatin (line 57-
72)?
Answer:
It is known that hydroxyproline and proline play a key role in collagen stability, in fact, they contribute to the formation and stability of the triple-helix structure of collagen due to the formation of hydrogen bonds. We pointed out this fact in the article (p. 4) and provided the relevant literature references [4,16,17,51]. A more detailed discussion of this issue was not intended in the article. We agree that in the Introduction this information is unnecessary and we deleted it from Introduction section.
- line 185 - 188. the authors quite boldly, actually without providing proof, suggest
“Apparently” that the presence of alginate and chitosan in the composition with fish gelatin
prevents the reversible rehabilitation of collagen-like triple helixes. On what basis was such a
conclusion made?
Answer:
We have provided this statement as a result of the analysis of powder diffraction data for gelatin composites with alginate and chitosan. It is for these samples that the peak responsible for the formation of collagen-like helix structures is practically not observed according to powder diffraction data. These facts are already presented in detail in the article (pp. 4-5) and they are actually the premise of our statement.
- line 278-187. reference should be made to more similar results, which will give more
credence to the authors' results obtained.
We suppose that gave enough references on outsider and own articles discussing the results.
- The section 2.4 SEM Visualization of Hydrogel Morphology, line 477-484. can the authors
refer to the influence of the structure of the different materials on the applicability?
Answer: No, we did not analyze our results from the point of detailed applicability. We only pointed the application fields where our results may be useful. The journal IJMS is not aimed on the application problems.
- line 553-555. the authors report that the content of K+, Ca2+ and Na+ ions in the
polysaccharide did not exceed 6, 1 and 1% by weight, respectively. What effect on the
polysaccharide can increasing the %wages of these ions have?
Answer: We agree with this comment of dear Reviewer. Indeed, we don’t use this information and it is the excessive one. Thus, we removed this information from the text.
- why were the studies described in Section 3.3 Small-Angle and Wide-Angle X-ray Scattering
performed at 1°, 26° and 45°C?
Answer: If 1C is the state of a gel, 45C is the state of a sol, and 26C is to see what state of the system will be closer to gel or sol. We used these temperatures to have a complete view on the difference in the structural state of studied systems.
- Complete the literature list with missing DOI numbers.
Answer: Done. Thank you for this comment.
For each graphic, please indicate the source - if it is not prepared by the authors of the
manuscript. What programs and licenses were used to prepare the graphics included in the
manuscript? To be completed.
Answer:
Every graphic and figure are prepared by authors of manuscript. We used the Microsoft Paint (commonly known as MS Paint or simply Paint) is a simple raster graphics editor that has been included with all versions of Microsoft Windows. Usually this information is not published.
Dear Reviewer, thank you for your time. We hope that these corrections will lead to the improvement of manuscript.
Round 2
Reviewer 2 Report
Comments and Suggestions for Authors
The authors have answered the Reviewer's questions.
No further changes to the manuscript are necessary.
It can be accepted for publication in IJMS.